# Influence of temporary emigration on wood turtle (*Glyptemys insculpta*) detectability, with implications for abundance estimation

**Allyson N. Beard**[1], **Donald J. Brown**[1,2]*, **Eric T. Hileman**[1], **Michael T. Jones**[3], **Jena M. Staggs**[1], **Ron A. Moen**[4], **Andrew F. Badje**[5], **Christopher M. Lituma**[1]

**1** School of Natural Resources, West Virginia University, Morgantown, West Virginia, United States of America, **2** Pacific Northwest Research Station, USDA Forest Service, Amboy, Washington, United States of America, **3** Massachusetts Division of Fisheries and Wildlife, Westborough, Massachusetts, United States of America, **4** Natural Resources Research Institute, University of Minnesota-Duluth, Duluth, Minnesota, United States of America, **5** Bureau of Natural Heritage Conservation, Wisconsin Department of Natural Resources, La Crosse, Wisconsin, United States of America

* Donald.Brown2@usda.gov

**Data Availability Statement:** All relevant data are within the manuscript and its Supporting information files.

## Abstract

Reliable population estimates are important for making informed management decisions about wildlife species. Standardized survey protocols have been developed for monitoring population trends of the wood turtle (*Glyptemys insculpta*), a semi-aquatic freshwater turtle species of conservation concern throughout its distribution in east-central North America. The protocols use repeated active search surveys of defined areas, allowing for estimation of survey-specific detection probability ($p$) and site-specific abundance. These protocols assume population closure within the survey area during the survey period, which is unlikely to be met as wood turtles are a highly mobile species. Additionally, current protocols use a single-pass design that does not allow for separation of availability ($p_a$) and detectability ($p_d$). If there are systematic influences on $p_a$ or $p_d$ that are not accounted for in the survey design or data analysis, then resulting abundance estimates could be biased. The objectives of this study were to determine if $p_a$ is a random process and if $p_a$ and $p_d$ are influenced by demographic characteristics. We modified the wood turtle survey protocol used in the upper Midwest to include a double-pass design, allowing us to estimate $p_a$ and $p_d$ using a robust design capture-recapture model. The modified protocol was implemented at 14 wood turtle monitoring sites in Minnesota and Wisconsin between 2017 and 2022. Our results indicated that $p_a$ was non-random and that $p_d$ increased with turtle carapace length. Our study suggests that model assumptions for current wood turtle population models may be violated, likely resulting in an overestimation of abundance. We discuss possible protocol and modeling modifications that could result in more accurate wood turtle abundance estimates.

## Introduction

Population monitoring programs are important to make informed decisions in wildlife management, allowing managers to track changes in occupancy and abundance and track

**Funding:** Funding for data collection was provided by the Minnesota Department of Natural Resources through a U.S. Fish and Wildlife Service Competitive State Wildlife Grant (#F14AP00028; RAM and DJB) and the Wisconsin Department of Natural Resources through a U.S. Fish and Wildlife Service Great Lakes Fish and Wildlife Restoration Act grant (#F21AP00170; AFB and DJB). Funding for data analysis was provided by the Pennsylvania Fish & Boat Commission through a U.S. Fish and Wildlife Service Competitive State Wildlife Grant (#4100091099; DJB and CML). The funders had no role in study design, data collection and analysis, decision to publish, or preparation of the manuscript.

**Competing interests:** The authors have declared that no competing interests exist.

population responses to management actions [1, 2]. For species of conservation concern, such as those listed in the United States or Canada as threatened or endangered at either the state/province or federal level, changes in the number and size of populations are the primary metrics used for assessing species status and recovery [3, 4]. Thus, much research has been devoted to development and assessment of survey protocols and statistical models to improve reliability of population inferences [e.g., 5–8].

Temporary emigration of individuals is a process that can influence abundance ($N$) estimates [9–11]. Temporary emigration can include the movement of individuals outside of survey areas as well as movement within survey areas that results in non-detectability for the sampling design. For example, portions of a terrestrial salamander population can be subterranean during a sampling event, making them unavailable for detection by surveyors [12]. Temporary emigration results in an individual being unavailable to be observed during a portion of the sampling events, which is separate from the probability of detecting the individual given that it is available. Given the presence of an individual in the sampling area at some point during the sampling period, detection ($p$) can be broken down into two components: availability ($p_a$) and detection given availability ($p_d$) [10]. However, monitoring programs rarely use population survey designs that provide the information needed to estimate both components [e.g., 13–15].

When separate estimates of $p_d$ and $p_a$ are not possible in populations that experience temporary emigration, $N$ refers to the superpopulation size (i.e., the number of individuals that are available for detection at any time during the survey period), with the implicit assumption that $p_d$ is constant and $p_a$ is random among individuals and surveys, or that model covariates can explain the variation. Superpopulation estimates should have minimal bias when these assumptions are met [16, 17]. However, if there are systemic influences on $p_d$ or $p_a$ that are not accounted for, such as inherent differences between sex or age groups due to morphology or behavior, then the resulting $N$ estimates could be biased [18].

The wood turtle (*Glyptemys insculpta*) is a semi-aquatic freshwater turtle and species of conservation concern endemic to east-central North America [19–22]. Unlike many small vertebrates, wood turtles have long life spans with high adult annual survivorship (typically >0.9) and delayed sexual maturity (11–20 years of age) [23]. This life history strategy is likely not conducive to a world with rapid anthropogenic change [24, 25]. Wood turtles are currently categorized as threatened by the Committee on the Status of Endangered Wildlife in Canada [26] and are under review for listing under the U.S. Endangered Species Act [27]. Recently, two standardized survey protocols were developed to monitor wood turtle population trends in the midwestern and northeastern United States [8, 25, 28, 29]. Both protocols consist of repeated surveys using a single pass through a designated stream segment, allowing for estimation of survey-specific $p$ and site-specific $N$. Because the protocols use a single-pass survey design, they do not allow for separation of survey-specific $p$ into the components of $p_a$ and $p_d$ [30].

Wood turtles typically remain near flowing water throughout the year [31–34]. However, they are largely terrestrial from late spring to early fall [35–37]. Results from studies that tracked the movement of individual wood turtles indicated survey areas are not likely to remain closed to population change during the survey time frames, as the turtles continuously move throughout stream, riparian, and upland habitats [32, 33, 37–39]. Currently, $N$-mixture models are the most common method used to estimate $N$ using wood turtle monitoring program data [e.g., 8, 25]. Key assumptions of $N$-mixture models include that all individuals have the same detection probability and that the population is closed, or availability is a random process during the sampling period [40]. Whether the temporary emigration of wood turtles during the survey period is a random or Markovian process (i.e., dependent on the previous state) has not been assessed [sensu 41]. Therefore, it is unclear if current $N$ estimates accurately represent the superpopulation size.

Our objective was to assess sampling assumptions for existing wood turtle survey protocols [8, 29], specifically to determine if $p_a$ is a random process and whether $p_a$ and $p_d$ are influenced by demographic characteristics. Previous studies that tracked locations of individual turtles suggest that movement and habitat use patterns differ between males and females [e.g., 33, 38, 42], and it is assumed that smaller turtles are harder to detect during surveys [43]. Thus, we hypothesized that $p_a$ would be influenced by turtle sex and $p_d$ would be influenced by turtle size. To accomplish these objectives, we modified the Midwest protocol to include a double-pass design, thus allowing us to separately estimate $p_a$ and $p_d$ using a robust design capture-recapture model. We used our survey results to assess support for the presence of temporary emigration and the influence of demographic characteristics on $p_a$ and $p_d$. The results of this study will assist with future refinements of the wood turtle population survey protocols to maximize reliability of population inferences.

## Methods

### Study area

We conducted this study at 14 wood turtle monitoring sites in the Laurentian Mixed Forest Province of Wisconsin and Minnesota, including 8 sites in northeastern Minnesota, 4 sites in northwestern Wisconsin, and 2 sites in northeastern Wisconsin (specific sampling locations withheld in compliance with Minnesota and Wisconsin data practices for species of conservation concern). Each monitoring site consisted of a 480–1,070-m stretch of river (mean = 704 m) and adjacent riparian and upland habitat. The study sites occurred in 4 HUC8 watersheds and were characterized by mixed hardwood and conifer forests with intermittent forest openings [44, 45].

### Data collection

We collected data over a 5-week period from early May to early June in northeastern Minnesota during 2017 and Wisconsin during 2021–2022. All data were collected using a survey design based on the Midwest standardized protocol, which consisted of 4 visual encounter surveys (VES) on foot at each site. Two observers surveyed each site by simultaneously searching for wood turtles along 2 transect bands on each side of the river, with transect centerlines placed at the river-land interface and inland at 15 m (Fig 1). Surveyors at the river-land interface wore polarized glasses to maximize detectability of wood turtles in the water near the river's edge [46]. We modified the protocol for this study to include one additional pass along each transect band by an independent observer. Immediately after completion of the first pass, the observers switched transect bands and conducted a second survey on the same side of the river. The same process was repeated for the other side of the river. This resulted in a double pass (i.e., a secondary survey period) for each transect band within each primary survey period, thus allowing the populations to be considered closed between passes and open between survey days.

For each new wood turtle detected, we recorded the survey band being searched and time of detection, obtained standard measurements (e.g., midline carapace length), marked the individual using carapace notches [47], photographed the individual, and released the individual where it was detected. For all within-year recaptures, we recorded locations but did not re-measure individuals. For each survey replicate, we recorded the date of survey, air temperature (˚C) at the beginning and end of the survey period, survey start and end time, and total search time (min). Capture and handling methods were approved by the Minnesota Department of Natural Resources, Wisconsin Department of Natural Resources, University of Minnesota

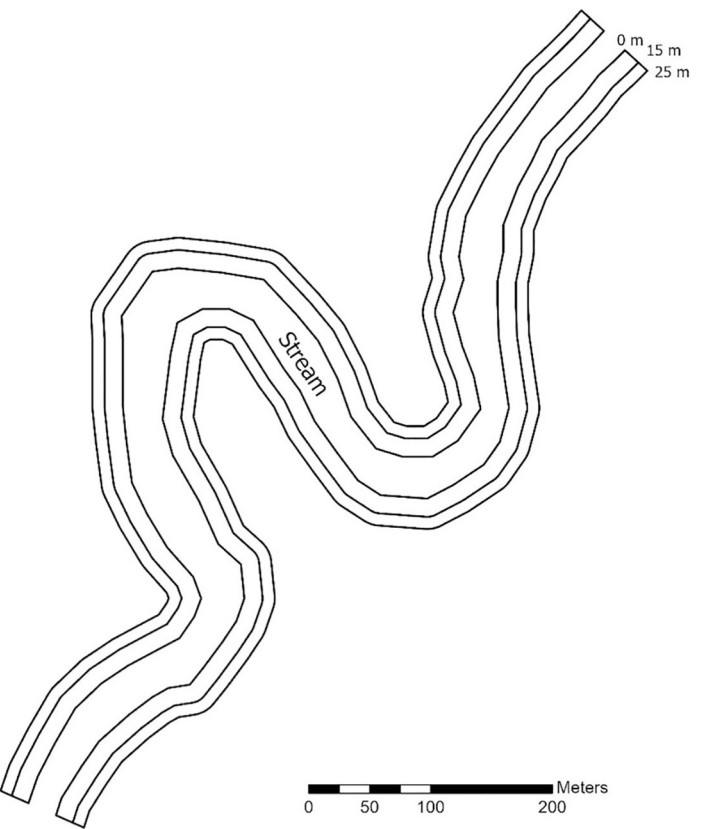

**Fig 1. Graphical representation of the sampling design used for wood turtle population surveys in Minnesota and Wisconsin.** We conducted visual encounter surveys (VES) along 2 transect bands on both sides of a stream segment, with transect centerlines placed at the river-land interface and inland 15 m. Transect bands included the area within 10 m of the center line, with an outer limit at 25 m from the stream. We conducted 2 passes along each transect band within each survey, resulting in a secondary survey period (i.e., double pass) for each transect within each primary survey period.

Institutional Animal Care and Use Committee (Protocol No. 1504-32514A), and West Virginia University Institutional Animal Care and Use Committee (Protocol No. 2002033297).

## Statistical analysis

We estimated capture probability ($p_{dc}$; $p$ in Program MARK), recapture probability ($p_{dr}$; $c$ in Program MARK), and availability ($p_a$; 1-$\gamma$ in Program MARK) using a robust design model with the Huggins' estimator (Table 1; model 'Robust Design Huggins' $p$ and $c$') [48]. The Huggins' estimator is conditional on only detected individuals, allowing for the inclusion of individual-level covariates (e.g., carapace length) [49, 50].

For all models, we specified separate estimation of $p_{dc}$ and $p_{dr}$, because it was clear that probability of initial capture was substantially higher than the probability of recapture within primary periods. Our qualitative field observations indicate that individuals often respond to being handled by moving into the stream, thus reducing their probability of being recaptured during the second pass. We also fixed survival probability ($S$) at 1 because we assumed adult survival probability was near 1 during the 5-week survey window. A previous study in the Midwest found that most wood turtle mortality events occurred after the survey period during June and July [51].

**Table 1. Definitions of robust design sub-models used to create 11 candidate models for wood turtles (*Glyptemys insculpta*) at 14 population monitoring sites in Minnesota and Wisconsin.**

| Probability Model | Definition |
|---|---|
| Survival | |
| 1. $S$ (1) | Survival is fixed at 1.0 |
| Temporary Emigration | |
| 1. Markovian | Availability for capture at time $t$ is linked to availability status at time $t$-1 ($\gamma' \neq \gamma''$) |
| 2. Random | Availability for capture at time $t$ is independent of availability status at time $t$-1 ($\gamma' = \gamma''$) |
| 3. Markovian–M | Availability of males at time $t$ is Markovian and availability of females and juveniles at time $t$ is random |
| 4. Markovian–F | Availability of females at time $t$ is Markovian and availability of males and juveniles at time $t$ is random |
| 5. Markovian–J | Availability of juveniles at time $t$ is Markovian and availability of females and males at time $t$ is random |
| Temporary Emigration Covariates | |
| 1. $\gamma''$ (age) $\gamma'$ (age) | Availability differs by age class (juvenile, adult); $\gamma''$ and $\gamma'$ differ from each other by an additive constant (Markovian) |
| 2. $\gamma''$ (age) $\gamma'$ () | Availability differs by age class (juvenile, adult); $\gamma''$ is equal to $\gamma'$ (Random) |
| 3. $\gamma''$ (sex) $\gamma'$ (sex) | Availability differs by sex (male, female, juvenile); $\gamma''$ and $\gamma'$ differ from each other by an additive constant (Markovian) |
| 4. $\gamma''$ (sex) $\gamma'$ () | Availability differs by sex (male, female, juvenile); $\gamma''$ is equal to $\gamma'$ (Random) |
| 5. $\gamma''$ (.) $\gamma'$ (.) | $\gamma''$ and $\gamma'$ differ from each other by an additive constant (Markovian) |
| 6. $\gamma''$ (.) $\gamma'$ () | $\gamma''$ is equal to $\gamma'$ (Random) |
| Capture and Recapture | |
| 1. $p_{dc}$ (.) $p_{dr}$ (.) | Capture probability differs from recapture probability by an additive constant |
| 2. $p_{dc}$ (CL) $p_{dr}$ (CL) | Capture probability differs from recapture probability by an additive constant; Capture and recapture probability both differ by carapace length |

$p_{dc}$, capture; $p_{dr}$, recapture; $\gamma$, temporary emigration (i.e., unavailability).

Temporary emigration is further divided into estimates of the probability an individual will remain unavailable at time $t$ if it was unavailable at time $t$-1 ($\gamma'$) and the probability an individual will be unavailable at time $t$ if it was available at time $t$-1 ($\gamma''$).

We defined a series of candidate model structures based on the study objectives and used a model selection approach with Akaike's Information Criterion corrected for small sample size (AIC$_c$) to assess support for the models. First, we constructed models to assess whether turtle size (i.e., carapace length) and age class (juvenile, adult) were strong predictors of $p_{dc}$ and $p_{dr}$, which differed from each other by an additive constant. We then used the most parsimonious model based on AIC$_c$ as the base model to assess support for a random or a non-random Markovian temporary emigration ($p_a$; $\gamma$ in Program MARK) structure. The Markovian temporary emigration structure assumed that availability for detection at time $t$– 1 influences availability at time $t$ ($\gamma' \neq \gamma''$ in program MARK; the probability of being outside the search area during the next survey is different for individuals inside the search area [$\gamma''$] compared to those already outside the search area during the current survey [$\gamma'$]), whereas the random model assumed that availability status at time $t$ was not influenced by the previous availability status ($\gamma' = \gamma''$ in program MARK; every individual in the superpopulation has the same probability of being outside the search area during the next survey). A model with a Markovian structure allowed for separate estimates of the probability an individual will remain unavailable at time $t$ if it was unavailable at time $t − 1$ ($\gamma'$) and the probability an individual will be unavailable at

time $t$ if it was available at time $t - 1(\gamma'')$ [41]. We evaluated models with sex and age class as covariates for both the random and Markovian temporary emigration structures. We considered individuals with a carapace length <170 mm to be juveniles [31]. We also assessed if there was support for a Markovian temporary emigration structure for only males, females, or juveniles (Table 1). The Fletcher's $\hat{c}$ value for the most general model without individual covariates was 1.002, so we did not correct for overdispersion. We constructed and ranked models using program MARK version 10.1 [52].

## Results

In total, 186 unique wood turtles were captured, including 86 individuals across the 8 Minnesota sites and 100 individuals across the 6 Wisconsin sites (S1 Dataset). The most parsimonious $p_{dc}$ and $p_{dr}$ model from the first set of candidate models included carapace length as a covariate for both parameters (AIC$_c$ = 910.479; $w_i$ = 0.622), with age class also receiving strong support ($\Delta$AIC$_c$ = 1.03; $w_i$ = 0.371). Carapace length was included in all models in the second candidate set. Both $p_{dc}$ and $p_{dr}$ increased with carapace length in the top model from the second candidate model set (Fig 2). The mean estimate for $p_{dc}$ was 0.53 (95% CI: 0.41–0.65), and the mean estimate for $p_{dr}$ was 0.25 (95% CI: 0.19–0.32).

The top 4 models of the second candidate model set included a Markovian temporary emigration structure (Table 2; $\Sigma w_i$ = 0.815). Both the sex and age class temporary emigration models had substantial support. In the top model, $\gamma''$ was higher for juveniles (0.78; 95% CI: 0.62–0.88) compared to adults (0.62; 95% CI: 0.51–0.71); $\gamma'$ was also higher for juveniles (0.89; 95% CI: 0.78–0.95) compared to adults (0.78; 95% CI: 0.67–0.87).

## Discussion

Our results indicate that detection given availability ($p_d$) is higher than detection probabilities ($p$) that have been previously reported using the Midwest protocol. Brown et al. [8] reported a maximum detectability of approximately 0.2 during ideal conditions using the original protocol. In contrast, we report a mean capture probability of 0.53. Survey-specific availability is likely a strong contributing factor to the lower detection probability estimates of Brown et al. [8]. Specifically, our top model predicts that the probability of an adult wood turtle remaining available (1 - $\gamma''$) or becoming available for detection (1 - $\gamma'$) between primary survey periods was 0.38 and 0.22, respectively.

Based on our top model from the first candidate model set, carapace length is a strong driver of $p_d$. In our top model overall, estimates of $p_d$ increase from 0.32 for the smallest recorded turtle size (57 mm) to 0.63 for the largest recorded turtle size (234 mm), supporting previous qualitative observations [43]. While there may be habitat use differences between adults and juveniles that influence $p_d$, larger turtles are more conspicuous and thus are more likely to be seen during surveys. Since most population estimates use methods that cannot separate $p_a$ and $p_d$, we recommend that adult turtles (carapace length > 170 mm) [31] be analyzed separately from juveniles when estimating population size. A key assumption of $N$-mixture models, the abundance estimation method typically used for analyses of wood turtle monitoring program data [8, 25], is that all individuals have the same detection probability (although we note that customization of $N$-mixture models to incorporate stage-specific parameter estimates is possible [53]). Our study demonstrated that this assumption was violated due to the dramatic increase in detectability with carapace length. This violation is likely introducing bias into both $p$ and $N$ estimates from $N$-mixture models [18]. Separating juveniles from adults during analysis would reduce this bias by decreasing the variation in detectability within these age groups.

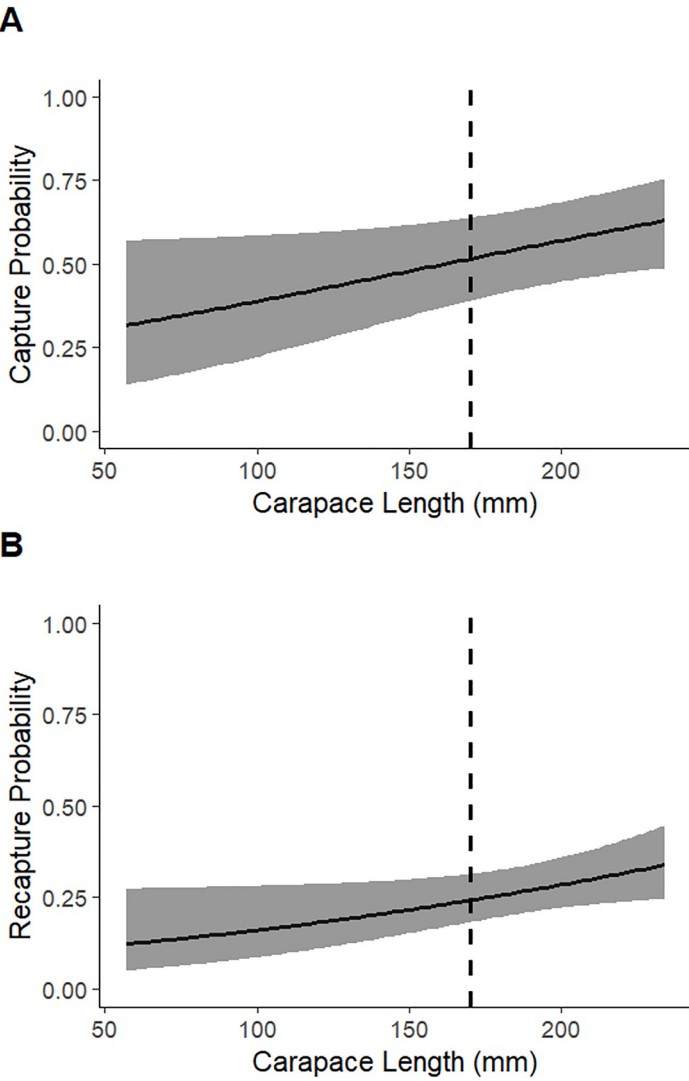

**Fig 2. Predicted capture (A) and recapture (B) probability of wood turtles (*Glyptemys insculpta*) in relation to midline carapace length.** Estimates based on population survey data collected between 2017 and 2022 at 14 monitoring sites Minnesota and Wisconsin. The dashed line represents the minimum carapace length used to classify an individual as an adult (170 mm). Gray shaded area represents the 95% confidence interval.

There was stronger evidence for Markovian movement than random movement throughout the survey period. This indicates that a turtle is more likely to remain outside the survey area once it has left, leading to an increased probability of unavailability as the survey period progresses. *N*-mixture models assume a closed population or random availability during the sampling period, and thus wood turtle movement dynamics are another potential source of bias in *p* estimation. Completing survey replicates close together in time, such as on consecutive days, might minimize this modeling bias. However, this might also come at the cost of eliminating the potential for detection of individuals that enter the site earlier or later than when the surveys are conducted, resulting in underestimation of the superpopulation size. Thus, retaining the typical survey strategy of replicating surveys over several weeks may be the optimal strategy, despite minor violation of model assumptions.

**Table 2. Candidate models for assessing temporary emigration probabilities of wood turtles (*Glyptemys insculpta*) at 14 monitoring sites in Minnesota and Wisconsin between 2017 and 2022.**

| Model Rank | Model[a] | K | AIC$_c$ | ΔAIC$_c$ | $w_i$ | Deviance |
|---|---|---|---|---|---|---|
| 1 | S (1) Markovian γ″ (age) γ′ (age) $p_{dc}$ (CL) $p_{dr}$ (CL) | 6 | 904.393 | 0.000 | 0.314 | 892.118 |
| 2 | S (1) Markovian γ″ (sex) γ′ (sex) $p_{dc}$ (CL) $p_{dr}$ (CL) | 7 | 904.473 | 0.080 | 0.301 | 890.104 |
| 3 | S (1) Markovian γ″ (.) γ′ (.)$p_{dc}$ (CL) $p_{dr}$ (CL) | 5 | 906.184 | 1.791 | 0.128 | 895.988 |
| 4 | S (1) Markovian—M γ″ (sex) γ′ (sex) $p_{dc}$ (CL) $p_{dr}$ (CL) | 7 | 907.333 | 2.939 | 0.072 | 892.964 |
| 5 | S (1) Random γ″ (sex) γ′ () $p_{dc}$ (CL) $p_{dr}$ (CL) | 6 | 908.102 | 3.709 | 0.049 | 895.827 |
| 6 | S (1) Random γ″ (age) γ′ () $p_{dc}$ (CL) $p_{dr}$ (CL) | 5 | 908.338 | 3.945 | 0.044 | 898.142 |
| 7 | S (1) Markovian—J γ″ (sex) γ′ (sex) $p_{dc}$ (CL) $p_{dr}$ (CL) | 7 | 908.430 | 4.037 | 0.042 | 894.062 |
| 8 | S (1) Markovian—F γ″ (sex) γ′ (sex) $p_{dc}$ (CL) $p_{dr}$ (CL) | 7 | 908.750 | 4.356 | 0.036 | 894.381 |
| 9 | S (1) Random γ″ (.) γ′ () $p_{dc}$ (CL) $p_{dr}$ (CL) | 4 | 910.479 | 6.086 | 0.015 | 902.349 |

$p_{dc}$, capture; $p_{dr}$, recapture; γ, temporary emigration (i.e., unavailability).

Temporary emigration is further divided into estimates of the probability an individual will remain unavailable at time *t* if it was unavailable at time *t*-1 (γ′) and the probability an individual will be unavailable at time *t* if it was available at time *t*-1 (γ″).

[a] Carapace length was identified as a strong predictor of capture ($p_{dc}$) and recapture (*c*) probability in preliminary models and was used as the base model for comparison.

In conclusion, we found that size and movement dynamics of wood turtles influence capture probabilities during standardized population surveys, with implications for model-derived abundance estimates. Our recommendation to separately estimate adult and juvenile abundances would likely improve accuracy of abundance estimates using *N*-mixture models. Unfortunately, there is no clear solution to address Markovian movement of wood turtles without doubling survey effort to explicitly account for survey-specific availability in models. We recommend that site-level movement dynamics during the standardized survey period be studied in other populations to improve our understanding of how common Markovian movement is among populations.

While our study focused specifically on investigating assumptions of a survey protocol for the wood turtle, many other species monitoring programs use count data from spatially constrained VES protocols to estimate occurrence or abundance, such as for eastern box turtles (*Terrapene carolina*) in Massachusetts [7], foothill yellow-legged frogs (*Rana boylii*) in California [54], and Shenandoah salamanders (*Plethodon shenandoah*) in Virginia [55]. Every species monitoring program operates with limited resources and the associated survey protocols reflect tradeoffs between intensity of sampling at individual sites and amount of spatial coverage across the monitoring area. While there are many different statistical frameworks and models available for VES data analyses, all models have assumptions and it is important to consider, and if needed experimentally test, whether the data being collected are appropriate for the anticipated statistical analysis. While it will not always be possible to modify protocols to meet every model assumption, in some cases minor modifications to the sampling protocol or statistical analyses could improve accuracy of the resulting model estimates, thus improving quality of the information used to guide species management decisions and actions.

## Supporting information

**S1 Dataset. Individual capture histories, sex and age classes, and carapace lengths, formatted for analysis in program MARK.**
(XLSX)

## Acknowledgments

We thank all the individuals who contributed suggestions for the original population survey protocol or assisted with population surveys for this study, particularly M. Berkeland and M. Cochrane. Any use of trade, product, or firm names is for descriptive purposes only and does not imply endorsement by the U.S. Government. The findings and conclusions in this publication are those of the authors and should not be construed to represent any official USDA of U.S. Government determination or policy.

## Author Contributions

**Conceptualization:** Donald J. Brown, Ron A. Moen.

**Data curation:** Allyson N. Beard, Donald J. Brown, Jena M. Staggs, Ron A. Moen.

**Formal analysis:** Allyson N. Beard, Donald J. Brown, Eric T. Hileman.

**Funding acquisition:** Donald J. Brown, Ron A. Moen.

**Investigation:** Allyson N. Beard, Jena M. Staggs, Ron A. Moen, Andrew F. Badje.

**Methodology:** Allyson N. Beard, Donald J. Brown, Michael T. Jones, Christopher M. Lituma.

**Project administration:** Donald J. Brown, Ron A. Moen, Christopher M. Lituma.

**Resources:** Donald J. Brown, Ron A. Moen, Andrew F. Badje.

**Supervision:** Donald J. Brown, Ron A. Moen, Andrew F. Badje.

**Visualization:** Allyson N. Beard.

**Writing – original draft:** Allyson N. Beard, Donald J. Brown.

**Writing – review & editing:** Eric T. Hileman, Michael T. Jones, Jena M. Staggs, Ron A. Moen, Andrew F. Badje, Christopher M. Lituma.

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
