## [Decision Letter · Decision Letter 0]

22 Jan 2024

PONE-D-23-42768Influence of temporary emigration on wood turtle (Glyptemys insculpta) detectability, with implications for abundance estimationPLOS ONE

Dear Dr. Brown,

Thank you for submitting your manuscript to PLOS ONE. After careful consideration, we feel that it has merit but does not fully meet PLOS ONE’s publication criteria as it currently stands. Therefore, we invite you to submit a revised version of the manuscript that addresses the points raised during the review process.

We look forward to receiving your revised manuscript.

Kind regards,

Karen Root, Ph.D.

Academic Editor

PLOS ONE

Journal Requirements:

2. Please note that PLOS ONE has specific guidelines on code sharing for submissions in which author-generated code underpins the findings in the manuscript. In these cases, all author-generated code must be made available without restrictions upon publication of the work. 

Please review our guidelines at https://journals.plos.org/plosone/s/materials-and-software-sharing#loc-sharing-code and ensure that your code is shared in a way that follows best practice and facilitates reproducibility and reuse.

5. We note that Figure 1 in your submission contain map images which may be copyrighted. All PLOS content is published under the Creative Commons Attribution License (CC BY 4.0), which means that the manuscript, images, and Supporting Information files will be freely available online, and any third party is permitted to access, download, copy, distribute, and use these materials in any way, even commercially, with proper attribution. For these reasons, we cannot publish previously copyrighted maps or satellite images created using proprietary data, such as Google software (Google Maps, Street View, and Earth). For more information, see our copyright guidelines: http://journals.plos.org/plosone/s/licenses-and-copyright.

(1) You may seek permission from the original copyright holder of Figure 1 to publish the content specifically under the CC BY 4.0 license.  

**Additional Editor Comments:**

This paper evaluates the potential influence of temporary movements on estimates of abundance of wood turtles. It will make an important contribution to our understanding of how to better monitor this highly mobile species, but the reviewers and I believe it needs a little more clarification of the approach used.  As reviewer 2 highlights, there are some potential biases that could be introduced using the double pass method, the implications of which should be explored further in the discussion.  For example, how does the increased intensity of searching affect the likelihood of movement of the turtles between the passes?  Reviewer 1 highlights a few areas that should be strengthened (with support and/or additional explanation) in both the introduction and the discussion. I think it would also be helpful to set up some expectation or predictions in the introduction to which the results could be compared.  For example, did you expect size to influence these parameters and why?  Similarly, reviewer 2 notes that the N-mixture model context may need to be discussed in the introduction as well the discussion.  Both reviewers have identified some specific areas that were unclear or need some revision.  This paper has a lot of potential but needs some revision before suitable for publication.

Reviewers' comments:

Reviewer's Responses to Questions

**Comments to the Author**

1. Is the manuscript technically sound, and do the data support the conclusions?

Reviewer #1: Yes

Reviewer #2: Partly

2. Has the statistical analysis been performed appropriately and rigorously? 

Reviewer #1: Yes

Reviewer #2: No

3. Have the authors made all data underlying the findings in their manuscript fully available?

Reviewer #1: Yes

Reviewer #2: Yes

4. Is the manuscript presented in an intelligible fashion and written in standard English?

Reviewer #1: Yes

Reviewer #2: Yes

5. Review Comments to the Author

Reviewer #1: Summary

Beard et al. re-evaluated turtle population models, indicating that detectability and availability must be systemically evaluated to avoid biased abundance models. The authors proposed separating adults and juveniles in these studies, to reduce detection biases.

General impression

The study is interesting and rigorous, and their results have immediate implications for turtle conservation.

Minor points

In the introduction it would be advisable for the authors to include references that support their hypothesis that pa and pd could be influenced by demographic traits (line 99).

The authors must explain what the Midwest protocol is, since Plos ONE is an international journal and most readers (included me) not know what this protocol consists of.

Ln 189. What has been the p previously reported in the Midwest protocol?

Ln 196-205. Authors should explain why they believe LCL alters the probability of detection, citing appropriate references.

In general, both the discussion and the intro are very focused on the target species, Glyptemys insculpta. Plos ONE is a generalist and not a herpetological journal, the authors should work on generalize their discussion and conclusions.

In the intro it would be necessary to introduce some peculiarities of the natural history of turtles, including their high inter-annual survival, since this is relevant to interpret the results of this study and totally differs from other small ectothermic vertebrates.

Reviewer #2: This is an interesting paper on modeling detection processes (availability and perception) in surveys of wood turtles. I think the problem is important because one has to properly understand these processes in order to interpret population size estimates and also for the design of effective surveys. I am generally favorable about the manuscript, although I worry that the double pass design is not really a double pass because there must be some memory in the observer, and also individual turtles might move. These should have different effects on the outcome, and maybe they cancel out, I don’t know, but it’s problematic!

Specific line-referenced comments:

line 35: the phrase "p_a is random" is used repeatedly in the paper,

but its meaning is unclear (and I believe it is used wrongly here).

The parameter p_a is a fixed parameter to be estimated, it is not a

"random parameter" nor is it even a random effect. I think you mean

to say that temporary emigration is random. That's quite different.

This needs to be explained somewhere.

line 58: the statement here that movement out of the survey area causes

individuals to be unavailable. In addition to that, individuals being

buried in the ground or water also make them unavailable, even if they

have not left the survey area.

line 60: after the current sentence ends at 'individual' you should add 'given that it is available."

line 64: detection given availability is often called perception (I think in some of the Nichols and related work).

line 69-70: sentence beginning "These estimates assume..." -- I just don't understand this. Some more detail and clarification is needed.

line 94: 'temporary emigration is random or directional' -- I think these

concepts need defined (or else I missed that perhaps).

line 99: here again is the statement that 'p_a is random' (see comment above)

Data collection: The protocol used here would be ideal to apply spatial

capture-recapture models since you have both spatial information about the search and the capture locations.

lines 128-130: this description of the double pass approach is not clear. Does this mean each transect was run twice? i.e., down and back? This is my understanding, but it is confusing because of "bands" and then repeated "for each transect" within "each primary". It's just not clear. Please work on the wording here and maybe add a hypothetical search line to Figure 2, in a different color.

In any case, about this double pass approach, I think this has to produce violations of basic assumptions. If you catch a turtle on the first pass, you are more likely to catch it on the 2nd pass (or perhaps less likely). There is some behavioral response of both the observer and the individual turtle. This has to be discussed and dealt with in some manner.

Certainly if the turtle did not move, the 2nd detection would not be stochastic but, rather, would occur with probability 1. (this is like a removal design in a sense).

line 150: insert "survival probability" after "fixed" and before "S"

line 160: I don't understand this gamma prime and gamma double prime notation. What does that mean? Why not use t and t-1 here? Define/explain please.

line 167: 'Markovian \\gamma structure' what does this mean?

line 176: "carapace length as a covariate" -- for both parameters? If so be explicit about that.

line 181: the phrase "Markovian gamma structure" is used here. You might as well just say "Markovian temporary emigration structure" because that is what is meant.

line 188: I think there is redundant notation: p_d and p_a are used to define parameters but also in the text the terms p and c are used to denote capture and recapture. How are these related to p_d and p_a ?

line 209: here the term 'directional movement' is used. This needs defined and related to previous concepts discussed in this paper.

The N-mixture model context comes up rather abruptly in the Discussion, and the paper is not about the N-mixture model so this is a little confusing. You might introduce the relevant context in the Introduction. Also, FWIW, the stratified N-mixture model is well suited for data collected by size class. One paper about this class of models is here:

Zipkin, E.F., Thorson, J.T., See, K., Lynch, H.J., Grant, E.H.C., Kanno, Y., Chandler, R.B., Letcher, B.H. and Royle, J.A., 2014. Modeling structured population dynamics using data from unmarked individuals. Ecology, 95(1), pp.22-29.

6. PLOS authors have the option to publish the peer review history of their article (what does this mean?). If published, this will include your full peer review and any attached files.

Reviewer #1: **Yes: **Daniel Escoriza

Reviewer #2: No

---

## [Author Response · Author response to Decision Letter 0]

22 Mar 2024

Detailed response to reviewers document included with resubmission.

---

## [Editor Report · Decision Letter 1]

28 Mar 2024

Influence of temporary emigration on wood turtle (Glyptemys insculpta) detectability, with implications for abundance estimation

PONE-D-23-42768R1

Dear Dr. Brown,

We’re pleased to inform you that your manuscript has been judged scientifically suitable for publication and will be formally accepted for publication once it meets all outstanding technical requirements.

Kind regards,

Karen Root, Ph.D.

Academic Editor

PLOS ONE

Additional Editor Comments (optional):

I appreciate the authors’ thoroughness and thoughtfulness in addressing the comments and suggestions by the reviewers.  With these revisions the paper is now suitable for publication.
---

## [Editor Report · Acceptance letter]

3 Apr 2024

PONE-D-23-42768R1 

PLOS ONE

Dear Dr. Brown, 

I'm pleased to inform you that your manuscript has been deemed suitable for publication in PLOS ONE. Congratulations! Your manuscript is now being handed over to our production team.

Kind regards, 

on behalf of

Professor Karen Root 

Academic Editor

PLOS ONE